Creating a Chinese suicide dictionary for identifying suicide risk on social media

Lv Meizhen 1 2
Li Ang 3 4 angli@bjfu.edu.cn angli1986@yahoo.com
Liu Tianli 5
Zhu Tingshao 1 6 tszhu@psych.ac.cn
1 Key Lab of Behavioral Science of Chinese Academy of Sciences (CAS), Institute of Psychology, CAS , Beijing , China
2 Department of Psychology, Peking University , Beijing , China
3 Department of Psychology, Beijing Forestry University , Beijing , China
4 Black Dog Institute, University of New South Wales , Sydney , Australia
5 Institute of Population Research, Peking University , Beijing , China
6 Key Lab of Intelligent Information Processing of CAS, Institute of Computing Technology, CAS , Beijing , China
Hochheiser Harry
Electronic publication date: 2015 Dec 15
Publication date: 2015
Volume: 3
Electronic Location ID: e1455
Received 2015 May 19; Accepted 2015 Nov 9
Copyright: © 2015 Lv et al.
Copyright year: 2015
Copyright holder: Lv et al.
License: This is an open access article distributed under the terms of the Creative Commons Attribution License, which permits unrestricted use, distribution, reproduction and adaptation in any medium and for any purpose provided that it is properly attributed. For attribution, the original author(s), title, publication source (PeerJ) and either DOI or URL of the article must be cited.
License URL: https://creativecommons.org/licenses/by/4.0/

Keywords: Weibo, Suicide risk, Microblog, Social media, China

Funding: National High-Tech R&D Program of China 2013AA01A606 National Basic Research Program of China 2014CB744600 Key Research Program of Chinese Academy of Sciences (CAS) KJZD-EWL04 CAS Strategic Priority Research Program XDA06030800 Fundamental Research Funds for the Central Universities NO. BLX2015-42 The authors were supported by the National High-Tech R&D Program of China (2013AA01A606), the National Basic Research Program of China (2014CB744600), the Key Research Program of Chinese Academy of Sciences (CAS) (KJZD-EWL04), the CAS Strategic Priority Research Program (XDA06030800), and the Fundamental Research Funds for the Central Universities (NO. BLX2015-42). The funders had no role in study design, data collection and analysis, decision to publish, or preparation of the manuscript.

==============================
Introduction. Suicide has become a serious worldwide epidemic. Early detection of individual suicide risk in population is important for reducing suicide rates. Traditional methods are ineffective in identifying suicide risk in time, suggesting a need for novel techniques. This paper proposes to detect suicide risk on social media using a Chinese suicide dictionary.

Methods. To build the Chinese suicide dictionary, eight researchers were recruited to select initial words from 4,653 posts published on Sina Weibo (the largest social media service provider in China) and two Chinese sentiment dictionaries (HowNet and NTUSD). Then, another three researchers were recruited to filter out irrelevant words. Finally, remaining words were further expanded using a corpus-based method. After building the Chinese suicide dictionary, we tested its performance in identifying suicide risk on Weibo. First, we made a comparison of the performance in both detecting suicidal expression in Weibo posts and evaluating individual levels of suicide risk between the dictionary-based identifications and the expert ratings. Second, to differentiate between individuals with high and non-high scores on self-rating measure of suicide risk (Suicidal Possibility Scale, SPS), we built Support Vector Machines (SVM) models on the Chinese suicide dictionary and the Simplified Chinese Linguistic Inquiry and Word Count (SCLIWC) program, respectively. After that, we made a comparison of the classification performance between two types of SVM models.

Results and Discussion. Dictionary-based identifications were significantly correlated with expert ratings in terms of both detecting suicidal expression (r = 0.507) and evaluating individual suicide risk (r = 0.455). For the differentiation between individuals with high and non-high scores on SPS, the Chinese suicide dictionary (t1: F1 = 0.48; t2: F1 = 0.56) produced a more accurate identification than SCLIWC (t1: F1 = 0.41; t2: F1 = 0.48) on different observation windows.

Conclusions. This paper confirms that, using social media, it is possible to implement real-time monitoring individual suicide risk in population. Results of this study may be useful to improve Chinese suicide prevention programs and may be insightful for other countries.

Introduction

Currently, suicide has been recognized as one of the most serious public health issue worldwide. According to World Health Organization, from 2000 to 2012, over 800,000 people in the world and 7.8 per 100,000 people in China died by suicide each year. Especially for people aged between 15 and 29 years, suicide is one of the leading causes of death (World Health Organization, 2014).

Early detection of suicide risk provides the basis for early intervention programs, which can be effective in preventing suicide deaths. However, in real life, suicidal people are not motivated to disclose their thoughts or plans before an attempt (World Health Organization, 2014), which requires to identify individuals at risk of suicide efficiently among populations. More importantly, individual suicide risk is associated with several risk factors (e.g., marital status and severity of depression), which do change over time (Brown et al., 2000; Roškar et al., 2011). Therefore, it is very important to identify suicide risk not only effectively but also timely.

Traditional methods (e.g., self-report ratings, structured interview, and clinical judgment) cannot identify individual suicide risk in real-time, which may lead to delayed reporting (McCarthy, 2010). For instance, for Web-based Injury Statistics Query and Reporting System (WISQARS) of Centers of Disease Control and Prevention in the United States (http://www.cdc.gov/injury/wisqars/index.html), the suicide data report delays almost 3 years.

The emergence of social media may shed light on this direction. Firstly, social media has a large user population. In China, the most popular Chinese microblogging service provider, Sina Weibo (weibo.com) has over 500 million registered users, producing more than 100 million microblogs per day. Particularly, there is a huge overlap between social media users and those with higher suicide risk. In China, 68% of Weibo users between 10 and 30 years in age (China Internet Network Information Center, 2014), which covers people with higher suicide risk (15–29 years) (World Health Organization, 2014). It suggests that social media may help us target a subset of the right people. Secondly, social media data is publicly available. All posts can be collected and processed in real time. Thirdly, social media data is informative. Social media users are motivated to discuss their health conditions online (Park, Cha & Cha, 2012; Prieto et al., 2014) and some individuals even have used social media to disclose their suicide thoughts and plans (Murano, 2014). In view of these advantages, it inspires us to identify individual suicide risk through social media analysis.

The words that people use provide important psychological cues to their mental health status (Rude, Gortner & Pennebaker, 2004; Jarrold et al., 2011). Many studies have found meaningful relationships between suicide risk and linguistic patterns in social media posts (McCarthy, 2010; Sueki, 2015), which suggests that linguistic features acquired from social media data can be used as indicators for identifying suicide risk. It means that an efficient detection of suicidal expression in social media posts is crucial to the identification of individuals with suicide risk among populations. Recently, some studies have built computational models for predicting suicide risk based on patterns of word use in social media posts (Paul & Dredze, 2014; O’Dea et al., 2015), which performed fairly well. However, since those words were selected based on expert knowledge, without a systematic framework, they are somehow limited and difficult to be expanded for improving the performance of computational models. Dictionary-based methods can be used to address this issue. Pestian et al. (2012) run a sentiment analysis on suicide notes using the Linguistic Inquiry and Word Count (LIWC) program (Pennebaker et al., 2007). Li et al. (2014a) used the Chinese Linguistic Inquiry and Word Count (CLIWC) program to conduct a case study on a suicidal user and analyzed all his blogs within one year before his suicide death. Huang et al. (2014) examined 53 suicidal users and explored linguistic features in their social media posts using a Chinese sentiment dictionary (HowNet). However, although previous studies confirm the validity of the dictionary-based method, dictionaries used in those studies are general-purpose programs for psycholinguistic analysis, which might be limited in detecting individual suicide risk more accurately. A dictionary is yet to be built for a particular purpose of identifying individual suicide risk on social media.

This study aims to build a Chinese suicide dictionary and test its performance in identifying individual suicide risk on social media.

Method

Our work consists of two steps: (1) Building the Chinese suicide dictionary and (2) Testing the performance of the Chinese suicide dictionary. Methods and procedures of this study have been approved by the Institutional Review Board of the Institute of Psychology, Chinese Academy of Sciences (the protocol number: H09036 and H15009).

Building the Chinese suicide dictionary

The Chinese suicide dictionary was built on Weibo using a four-step procedure: (1) Collecting Weibo posts; (2) Selecting initial words; (3) Filtering out irrelevant words; and (4) Expanding remaining words (see Fig. 1).

1. Collecting Weibo posts. To create a suicide dictionary, we need to find those Weibo users at risk of suicide and then examine the suicidal expression in their Weibo posts. To find out those Weibo users at risk of suicide, we contacted with Weibo user (逝者如斯夫 dead), who is famous for collecting relevant news reports and expressing his condolence on the death of those suicidal Weibo users. He provided us with a list of suicidal Weibo users. We confirmed the list by checking relevant news reports and looking through comments left on those suicidal Weibo accounts. Then, we conducted a further scrutiny of those confirmed users to exclude the following: (a) users who are not Chinese citizens (excluding one user); (b) users who update posts for business purposes (excluding two users); (c) users who updated less than 20 posts (excluding one user). Finally, we got a total of 31 suicidal Weibo users (12 males and 19 females) and downloaded their Weibo posts since registration.

Because a small number of suicidal users can be limited in exploring suicidal expression, we further randomly selected 1,000 regular users (368 men, 632 women, and 23.65 ± 5.935 years old) for examining their expression. All expanded users were selected from a customized Weibo database composed of 1.06 million active Weibo users (Li et al., 2014b). For each expanded user, we downloaded his/her three latest Weibo posts.

Finally, we acquired a total of 4,653 Weibo posts, including 1,653 Weibo posts from 31 suicidal users and 3,000 Weibo posts (1,000 × 3 = 3,000) from 1,000 expanded users.

Figure 1 Procedures in building the Chinese suicide dictionary.

2. Selecting initial words. Eight researchers (postgraduate students specializing in suicide research) were recruited to select Weibo posts with suicide risk using a framework of Rudd’s 12 warning signs for suicide risk (Rudd et al., 2006). After a training session, 50 in 4,653 Weibo posts were randomly selected and coded by eight coders independently with a good inter-coder reliability (α = 0.819). Then, the eight researchers were divided equally into two groups. Each one group coded one half of all Weibo posts and selected Weibo posts with suicide risk. In addition, each group was also instructed to pick up any word indicating suicide risk from those selected Weibo posts. Both the selection of suicidal Weibo posts and suicidal words were confirmed with an agreement of at least three coders in one group.

Moreover, because people at risk of suicide express negative emotions frequently (Li et al., 2012; Pestian et al., 2012; Li et al., 2014a), the same eight researchers were further instructed to select suicidal words from two Chinese sentiment dictionaries, HowNet (www.keenage.com) and NTUSD (Ku & Chen, 2007), using the same method.

Finally, we got a total of 7,908 initial words and then estimated the frequency of each word in the customized Weibo database for further analysis.

3. Filtering out irrelevant words. After collecting 7,908 initial words, we filtered out irrelevant words and categorized remaining words. To do so, another three researchers, who were also postgraduate students specializing in suicide research, were recruited to tune up the initial words. After a training session, they were instructed to filter out any word as follows: (a) words that change their meanings in different contexts (e.g., individuals suffering from discrimination might express their negative emotions using a word “unfair,” which can be recognized as a warning sign for suicide risk; while, the witness might express their sympathy using the same word, which cannot be recognized as a sign anymore); (b) words that are less sensitive to detect suicide risk; (c) words that appear in the customized Weibo database with a low frequency. If two of three coders agree to filter out one word, it would be eliminated. Upon a scrutiny of words, 1,862 words were kept in a preliminary suicide dictionary. Then, the same three researchers were further required to give weights to those remaining words, ranging from 1 (light) to 3 (heavy). Words with heavier weights are thought to be more sensitive to detect suicide risk. The weights of words were confirmed with an agreement of at least two coders. Among 1,862 words, 990 words were weighted as 1; 505 were weighted as 2; and 367 were weighted as 3.

After that, based on previous studies examining suicidal factors and suicidal themes (Phillips et al., 2002; Brezo, Paris & Turecki, 2006; World Health Organization, 2014; Jashinsky et al., 2014), we analyzed 1,862 words inductively, and developed an initial framework for categorizing those words. Then, the same three experts provided feedback on the framework, and a final framework was constructed (see Table 1). Using the framework, the three experts classified 1,862 words into 13 different categories with an agreement of at least two coders.

Table 1 Outline of the Chinese suicide dictionary.

Category	Number of words	Definition	Representative words	
Suicide ideation	586	Words reflecting suicidal thoughts	want to die (想死) escape (逃离)	
Suicide behavior	88	Words reflecting self-harm behaviors	seppuku (切腹) hypnotics (安眠药)	
Psychache	403	Words reflecting psychological distress	want to cry (想哭) loneliness (孤单)	
Mental illness	48	Words reflecting poor mental health status	depression (抑郁) hallucination (幻觉)	
Hopeless	188	Words reflecting a feeling of despair	dead end (死胡同) despair (绝望)	
Somatic complaints	183	Words reflecting somatic symptoms	headache (头疼) shortness of breath (透不过气)	
Self-regulation	36	Words reflecting an attempt to push oneself hardly	repression (压抑) force oneself to smile (强颜欢笑)	
Personality	72	Words reflecting negative personality	inferiority complex (自卑) hate oneself (讨厌自己)	
Stress	83	Words reflecting pressure in daily life	failure (输) pressure (压力)	
Trauma/hurt	182	Words reflecting traumatic or unpleasant experiences	get dumped (失恋) infidelity (出轨)	
Talk about others	47	Words reflecting one’s relatives and friends	partner (妻子) son (儿子)	
Shame/guilt	72	Words reflecting a feeling of shame and guilt	lose status (丢脸) making an apology (赔罪)	
Anger/hostility	180	Words reflecting a feeling of angry and hostile against others	damn it (他妈的) curse (诅咒)	

4. Expanding remaining words. Because of a frequent use of new words and phrases on social media, to improve the performance of a suicide dictionary in detecting innovative suicidal expression, we need to expand the suicide dictionary at all time. In this study, we developed a corpus-based method for expanding words automatically. Specifically, the suicide dictionary can be defined as a set of words (W) W=w,c,xi=1m

where w refers to each word in the suicide dictionary with its category (c) and weight (x). For each word (w), we search a corpus (C) for similar words (Ww) based on semantic similarity (F). The search process can be defined as w+C⟶FWw.

The semantic similarity between different words is estimated by word2vec, an open-source tool for computing vector representations of words (http://code.google.com/p/word2vec/). The performance of word2vec has been confirmed in previous studies (Mikolov et al., 2013).

We randomly selected Weibo posts with a capacity of 200 GB from a customized Weibo database (Li et al., 2014b) as the corpus and utilized Chinese Language Technology Platform (LTP) (Che, Li & Liu, 2010) for word segmentation. Using the word2vec, we searched the corpus for similar words. In this study, for each word, we only selected its four most similar words, which share the same category and weight.

Further, the same three researchers were also instructed to filter out irrelevant expanded words using the same criteria as mentioned in the section of “Filtering out irrelevant words.” If two of three coders agree to filter out one word, it would be eliminated. Therefore, we got a total of 306 expanded words. Both the category and the weight of expanded words might be tuned up by these three researchers with an agreement of at least two coders.

Finally, the suicide dictionary is composed of 2,168 words (1,862 + 306 = 2,168), which fit into 13 different categories.

Testing the Chinese suicide dictionary

After building the Chinese suicide dictionary, we tested its performance in identifying suicide risk on Weibo.

Participants

We broadcasted participant invitation on Weibo. A total of 1,196 Weibo users agreed to participate in this study. All participants were instructed to complete an online questionnaire and allow us to download their Weibo posts. From May 22th to July 13th in 2014, we received a total of 1,040 completed questionnaires. Among them, 252 participants were excluded based on the following criteria: (a) users who were less than 18 years old; (b) users who published less than 100 Weibo posts; (c) users who provided invalid answers on the online questionnaire; (d) users who had multiple user accounts (different accounts share the same IP address). Finally, a total of 788 participants were recognized as valid participants in this study (298 men, 490 women, and 24.23 ± 4.912 years old).

Measurement

Expert ratings were used as one of the two gold standards for evaluating the accuracy of the Chinese suicide dictionary in identifying suicide risk on social media.

Suicidal Possibility Scale (SPS) was used to measure individual levels of suicide risk (Cull & Gill, 1988), which can be recognized as another gold standard for evaluating the performance of the Chinese suicide dictionary. SPS is an effective screening tool designed to assess suicide risk in adolescents and adults (Gençöz, 2006; Naud & Daigle, 2010). SPS consists of 36 self-rating items. Participants rated themselves on each item by a 4-point Likert Scale (1 = None or little of the time to 4 = All of the time). We computed the Suicide Probability Score for each participant. High scores indicate high suicide risk. The Chinese version of SPS was used in this study (Liang & Yang, 2010). The Cronbach’s Alphas for the whole-questionnaire was 0.749 in our data. All 788 participants were divided into high risk and non-high (medium to low) risk group based on the distribution of SPS scores (69.35 ± 11.66). In other words, participants from high risk group of SPS scored more than 81.01 (69.35 + 11.66 = 81.01) and participants from non-high risk group of SPS scored less than 81.01.

Data analysis

We tested the performance of the Chinese suicide dictionary at the level of Weibo posts and Weibo users, respectively.

In terms of the analysis at the level of Weibo posts, we tested the performance in (a) detecting suicidal expression in Weibo posts, which can provide the basis for identifying individual suicide risk on social media.

As to the analysis at the level of Weibo users, we tested the performance of the Chinese suicide dictionary in (b) evaluating levels of individual suicide risk. Furthermore, we also tested its performance in (c) differentiating between individuals with high and non-high scores on SPS.

(a) Detecting suicidal expression in Weibo posts. For testing the accuracy in detecting suicidal expression, we compared dictionary-based identifications with expert ratings (Bantum & Owen, 2009).

We randomly selected users from both high and non-high risk group of SPS. In this study, we selected 30 users from each one group and got a total of 60 users (20 men, 40 women, and 24.43 ± 3.652 years old). The performance in detecting suicidal expression was tested on Weibo posts acquired from such 60 Weibo users. Repeated comparisons were made based on Weibo posts with different observation windows (i.e., 1 week, 1 month and 3 months before starting this study). The selection of observation windows depends on whether there are enough Weibo posts to be analyzed.

Specifically, for expert ratings, we recruited another three researchers specializing in suicide research for rating Weibo posts. All three coders were required to rate how each Weibo post relates to each one of the 13 categories defined in the Chinese suicide dictionary by a 7-point Likert-type scale (1 = Extremely low consistency to 7 = Extremely high consistency). For each Weibo post, the overall rating was acquired by aggregating ratings on each category. Then, the final expert ratings were the average ratings of all three coders. For dictionary-based identifications, both the overall frequency of all dictionary words and the frequency of particular words within each category were estimated by matching the words in each Weibo post automatically.

In psychological studies, to validate a new measuring tool, it needs to test correlations between the new tool and a popular tool measuring the same psychological feature, which is known as convergent validity. If the two measures that theoretically should be related are in fact related, the new tool should be assumed as a valid one. Because the expert-rating method is commonly used to detect suicide risk before (McCarthy, 2010), in this study, we run correlations between dictionary-based identifications and expert ratings.

(b) Evaluating levels of individual suicide risk. For testing the accuracy in evaluating levels of individual suicide risk, we compared dictionary-based identifications with expert ratings.

In this study, we tested the convergent validity by estimating correlations between both the two measures. The performance in evaluating levels of individual suicide risk was tested on the same 60 Weibo users as mentioned in the section (a) (30 users from high risk group of SPS and 30 users from non-high risk group of SPS).

Specifically, for expert ratings, the same three researchers were instructed to evaluate individual levels of suicide risk by reading through all his/her Weibo posts. Each coder was required to rate individual suicide risk by a 7-point Likert-type scale (1 = Extremely low risk to 7 = Extremely high risk). Then, the final expert ratings were the average ratings of all three coders. For dictionary-based identifications, the frequency of dictionary words in each post were counted, and the weight of those dictionary words found in each post were summed up. If the total score of one post is up to three, it is recognized as the one with suicide risk. For each user, the proportion of Weibo posts with suicide risk is considered as his/her levels of suicide risk, which is used to compare with expert ratings.

(c) Differentiating between individuals with high and non-high scores on SPS. For testing the accuracy in differentiating between individuals with high and non-high scores on SPS, we built Support Vector Machines (SVM) models (Cortes & Vapnik, 1995) on the Chinese suicide dictionary and the Simplified Chinese Linguistic Inquiry and Word Count (SCLIWC) program (Gao et al., 2013), respectively. The classification accuracy of the SCLIWC models can be recognized as the baseline.

Models were built on all 788 participants who completed SPS successfully, and evaluated on different observation windows (i.e., 1 month and 2 months before conducting SPS test). The selection of observation windows depends on whether there are enough Weibo posts to be analyzed. To build SVM models, we extracted a feature vector (X) from all Weibo posts of each user. Elements (X1, X2……Xn) in this feature vector represent the ratio of words in different categories, which are defined by either the Chinese suicide dictionary or the SCLIWC. We compared the classification performance between two types of SVM models on different observation windows.

Results

Detection of suicidal expression in Weibo posts

Table 2 presents the test results. With a 1-week observation window, for the estimation of suicidal expression in 13 different categories, 11 of 13 correlation coefficients were significant between dictionary-based identifications and expert ratings, ranging from 0.263 to 0.913. The correlations on both of personality and trauma/hurt were not significant. For the overall estimation of suicidal expression, the correlation coefficient was 0.507 (p < 0.01). These correlations decreased with an increase in length of observation window.

Table 2 Comparison of the performance in detecting suicidal expression between dictionary-based identifications and expert ratings.

Observation window	Correlations between dictionary-based identifications and expert ratings	
	SI	SB	Psy	MI	H	SC	SR	Pers	S	T/H	TAO	S/G	A/H	O	
1 week	.651**	.750**	.551**	.459**	.406**	.913**	.400**	.047	.480**	.043	.329*	.901**	.263*	.507**	
1 month	.254*	.146	.437**	.032	.077	.637**	.300*	.291*	.328*	.027	.138	−0.007	.300*	.188	
3 months	.289*	.105	.485**	−0.046	.182	.227	.124	−0.029	−0.014	−0.037	0.050	.436**	.271*	.063	
Notes.

N = 60.

SI Suicide ideation

SB Suicide behavior

Psy Psychache

MI Mental illness

H Hopeless

SC Somatic complaints

SR Self-regulation

Per Personality

S Stress

T/H trauma/hurt

TAO Talk about others

S/G Shame/guilt

A/H Anger/hostility

O Overall estimation

* p < 0.05.

** p < 0.01.

Evaluation of levels of individual suicide risk

Results showed that, for evaluating levels of individual suicide risk, the correlation coefficient between dictionary-based identifications and expert ratings was 0.455 (p < 0.01).

Differentiation between individuals with high and non-high scores on SPS

Table 3 presents the test results. For the same observation window, the Chinese suicide dictionary (t1: F1 = 0.48; t2: F1 = 0.56) performs better than SCLIWC (t1: F1 = 0.41; t2: F1 = 0.48).

Table 3 Predicting high vs. non-high risk group of SPS using the Chinese suicide dictionary and the SCLIWC.

	Precision	Recall	F-measure	
1 month				
Suicide dictionary	0.60	0.40	0.48	
SCLIWC	0.43	0.40	0.41	
2 months				
Suicide dictionary	0.49	0.64	0.56	
SCLIWC	0.48	0.48	0.48	
Notes.

N = 788.

Precision is the fraction of retrieved instances that are relevant.

Recall is the fraction of relevant instances that are retrieved.

F-Measure is the harmonic mean of precision and recall.

Discussion

This study built a Chinese suicide dictionary for identifying suicide risk on social media and tested its performance on Chinese social media (Sina Weibo). This study confirmed that a real-time monitoring of suicide risk in population can be realized through social media analysis.

Firstly, the Chinese suicide dictionary can be used to detect suicidal expression in social media posts. For an overall estimation, a moderate correlation existed between dictionary-based identifications and expert ratings (r = 0.507), suggesting an acceptable level of convergent validity (Rogers, Lewis & Subich, 2002; Posner et al., 2011). However, correlations were not significant on categories of personality and trauma/hurt. It might be because, for human coders, some aspects of individual personality cannot be estimated easily through social media analysis (Qiu et al., 2012; Li et al., 2014b), which might lead to a decrease in correlations between dictionary-based identifications and expert ratings. Besides, Holmes et al. (2007) concluded that traumatic experience is tightly associated with negative emotional expressions and feelings of physical pain. It means that words which should be included in the category of trauma/hurt might be actually assigned to other similar categories (e.g., psychache, hopeless, somatic complaints), which leads to a non-significant correlation on trauma/hurt between two different measures. More importantly, with an increase length of observation time, the performance of the suicide dictionary declines. It might be due to the increase number of innovative words and phrases in Weibo posts which have not been included in the suicide dictionary yet. Because the Chinese suicide dictionary itself can be updated automatically, the performance can be improved in the future on any new corpus.

Secondly, based on the accurate detection of suicidal expression in Weibo posts, the Chinese suicide dictionary can be used to evaluate levels of individual suicide risk. A moderate correlation existed between dictionary measures and expert ratings (r = 0.455), suggesting an acceptable level of convergent validity.

Thirdly, apart from evaluating levels of individual suicide risk, the Chinese suicide dictionary also can be used to differentiate between individuals with high and non-high scores on self-rating measure of suicide risk. The Chinese suicide dictionary produces a more accurate estimation than the general-purpose dictionary for psycholinguistic analysis (e.g., SCLIWC). It means that building a dictionary for a particular purpose of identifying suicide risk on social media is worthwhile.

It is important to note the limitations of this study: (a) The sample size is a bit limited. We tested the performance in both detecting suicidal expression and evaluating levels of individual suicide risk on Weibo posts acquired from only 60 participants. Collecting and analyzing posts from a larger number of social media users might further validate the performance of the Chinese suicide dictionary; (b) The Chinese suicide dictionary was built and tested on Sina Weibo. We are not sure whether it will perform well on other Chinese social media platforms; (c) This study excluded low-frequency words in the Chinese suicide dictionary, which might also be sensitive to the variation of suicide risk; (d) The development of the suicide dictionary is based on a closed-vocabulary approach, which might limit findings to preconceived relationships with words or categories (Schwartz et al., 2013). Extracting a data-driven collection of words might further improve the performance in identifying suicide risk; (e) The Chinese suicide dictionary focuses on the frequency of words in a particular category and does not take contextual factors into account. Changes in elements of surrounding linguistic context may determine the relationship between individual psychological features and patterns of language use (Jarrold et al., 2011). Future works should not only focus on the frequency of words, but also examine vocabulary words in context; (f) In the Chinese suicide dictionary, the weights of words were assigned by experts. We are not sure whether the automated weight assignment techniques could improve the performance of the Chinese suicide dictionary; (g) The suicide dictionary is built for identifying suicide risk, which cannot predict the suicide action.

However, this study provides an innovative framework to prevent suicide in an effective manner. That is, through social media analysis, we can monitor individual suicide risk and capture those at high risk of suicide. After that, we can deliver intervention programs to those people immediately, which will be beneficial to improve the performance of suicide prevention.

Conclusion

This paper built a Chinese suicide dictionary to detect suicide risk on social media. Results indicate that the Chinese suicide dictionary works fairly well in identifying suicide risk at both levels of posts and users. The Chinese suicide dictionary can be used to implement real-time monitoring of suicide risk in population, thus improving the performance of suicide prevention and mental health promotion.

Supplemental Information

Supplemental Information 1 Raw data

Click here for additional data file.

Additional Information and Declarations

Competing Interests

Author Contributions

Human Ethics

Data Availability

The authors declare there are no competing interests.

Meizhen Lv conceived and designed the experiments, analyzed the data, contributed reagents/materials/analysis tools, wrote the paper, prepared figures and/or tables.

Ang Li conceived and designed the experiments, performed the experiments, wrote the paper, prepared figures and/or tables, reviewed drafts of the paper.

Tianli Liu conceived and designed the experiments, wrote the paper, reviewed drafts of the paper.

Tingshao Zhu conceived and designed the experiments, performed the experiments, analyzed the data, contributed reagents/materials/analysis tools, wrote the paper, performed the computation work, reviewed drafts of the paper.

The following information was supplied relating to ethical approvals (i.e., approving body and any reference numbers):

The Institutional Review Board of the Institute of Psychology, Chinese Academy of Sciences (the protocol number: H09036 and H15009).

The following information was supplied regarding data availability:

Raw data is available in the Supplemental Information.

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
