# Peer review of "Creating a Chinese suicide dictionary for identifying suicide risk on social media"

_PeerJ, doi:10.7717/peerj.1455_

## Round 0.1 · original submission · Major Revisions

· Academic Editor

Major Revisions

The reviewers have identified some significant issues with this paper. Having read the paper and the reviews, I believe that their concerns are valid and appropriate. I would like to particularly draw the authors' attention to the need for careful rewriting of the paper, ideally with help from a writer with experience in editing English text.

Additional details in the description of methods would be most helpful. Details such as the nature of the "experts" were not provided, leaving questions as to the validity of the approach.

Please also address two issues regarding Table 3 - I was confused by the claim that "dictionary-based models performed better in predicting one’s scores on SPS than LIWC-based models on different observation window". From my read of table 3, it looks as if the predictions were only better within the same month. Furthermore, I think there may be some problems with the F1 calculations in Table 3. Please recheck those numbers.

Reviewer 1 ·

Basic reporting

1. I do not understand the meaning of Figure 2, possibly because I am not familiar with computer science .
2. Conventionally, r = 0.4–0.5 is not “strong.” I think this is a “moderate” correlation. If you include interpretations anywhere in the paper, it is preferable not to include them in the Results section.
3. It is necessary to include a description that the suicide dictionary cannot predict the occurrence of suicide as a limitation in this study. In the future study , I hope that this problem will be solved.
4. You would be better off making a concrete suggestion to prevent suicide based on your suicide dictionary in the Discussion.

Experimental design

No Comments

Validity of the findings

No Comments

Additional comments

No Comments

Reviewer 2 ·

Basic reporting

No comments

Experimental design

No comments

Validity of the findings

No comments

Additional comments

This paper developed a suicide dictionary from microblogs and compared performance of the suicide dictionary with LIWC. The idea is meaningful and interesting in the field of language analysis. However, the developing procedure for the suicide dictionary and the result sections are not clear and comprehensive enough, causing some major and minor issues that should be dealt with first before this paper is considered to be accepted. The following points are my major concerns.

The development of the suicide dictionary

(1) The authors consulted a famous Weibo user ‘逝者如斯夫 dead’, who collects a lot of suicide posts. I am curious about the motivation of this user. Why does he collect those suicide posts? How does he collect those information from internet? Does he use some keywords to search suicide posts? If so, what are those keywords? Collecting suicide posts doesn’t make the person become an expert on the issue of suicide. If he is just a lay person who is interested in collecting suicide posts, why did the authors take his suggestion? The background about the Weibo user ‘逝者如斯夫 dead’ should be addressed more.

(2) The authors said they took suggestions from the Weibo user ‘逝者如斯夫 dead’. What are his suggestions? When he gave suggestions, what information source did he rely on? How do we know that Weibo user’s suggection is reliable and trustworthy?

(3) On line 140: Eight coders selected keywords from Weibo posts. What is the criteria they use to select keywords? Did the eight coders receive any professional training for selecting keywords? If so, the authors should describe the training process.

(4) On line 163: Three experts assigned weight coefficients (from 1 to 3) to remaining words. What is the three experts’ performance? What is the internal consistency among the three experts on weight coefficients?

(5) In table 1, the authors classified 2,168 words into 13 different categories without addressing how they did it and who did it. It’s important to understand how the authors classified those words.

(6) In table 2, the correlation between expert-rating and dictionary-rating on variables personality and trauma/hurt were not significant. Why? The authors should address this non-significant result.

(7) The bubble diagram looked fancy but provided no useful information. The authors may consider remove that section.

(8) The most important analysis in this paper is to use self-made suicide dictionary and LIWC dictionary to predict SPS and see which one has better prediction performance. In authors’ analysis, I didn’t see any detail information about LIWC-based model and suicide dictionary-based model. How did the authors establish the two models? Did the author use regression analysis to compare LIWC-based model and suicide dictionary-based model? If so, they author should clearly address that. Also, the authors claimed that the self-made suicide dictionary performed better than LIWC dictionary on predicting SPS. I didn’t see any statistical test for this result. The authors should take a statistical test to compare regression coefficients between the two models for their conclusion.

If the authors could respond to those above questions appropriately, I would love to see PeerJ accepts this interesting paper.

Reviewer 3 ·

Basic reporting

There are a large number of expository writing issues with this paper.
I recommend the authors find a good native English speaker to help
with the editing. If the author's search this review for quotes from
the article and find [sic] that is a pointer to where a there is some
kind of wording problem needing to be fixed. The number of [sic]'s is
an indication of the frequency and severity of the expository writing
issues.

I am a bit concerned with the author's idiosyncratic usage of basic
relevant terminology. For example, in the abstract, "the suicide
dictionary performs as efficiently as human coding in detecting
suicidal expression [sic] and identifying one's level of suicide
risk". 'Efficiency' does not seem to be correct metric here - it is a
measure of work done per unit of energy. Here are some metrics that
would be more appropriate: 'performance', 'accuracy', 'F measure',
'classification performance', 'sensitivity/specificity',
'recall/precision'. 'Efficiency' is just not the right term for use
in a CS-focused journal.

Another example raising this concern is found in paragraph 2 of the
Results, "The sensitivity was evaluated on each of 60 Weibo user
[sic]..." The use of the term 'sensitivity' seems idiosyncratic. Why
did not they say "accuracy" or "predictiveness" or "classification
performance" etc. I wonder if the authors realize that 'sensitivity'
is a technical term that is synonymous with 'recall'? (Note the terms
'sensitivity/specificity' used in medical journals which are related
to 'recall/precision' which are familiar to computer scientists).

Some more minor points:

The authors should be aware of the following article and may wish to
cite it. This is because I believe it provides another example of a classification system that is "better than LIWC". It also may be useful to describe whether your system is an Open or Closed Vocabulary approach.

Personality, Gender, and Age in the Language of Social Media: The
Open-Vocabulary Approach

http://journals.plos.org/plosone/article?id=10.1371/journal.pone.0073791

"We replicate traditional language analyses by applying Linguistic
Inquiry and Word Count (LIWC) [11], a popular tool in psychology, to
our data set. Then, we show that open-vocabulary analyses can yield
additional insights (correlations between personality and behavior as
manifest through language) and more information (as measured through predictive accuracy) than traditional a priori word-category
approaches."

Experimental design

One of the biggest problems with this paper is they do not give us
sufficient details about the machine learning approach - what
algorithm(s) were tried? what kind of feature selection was used? I
am even unclear on how the training set(s) and the test set(s) were
constructed. There was not much about machine learning approach aside
from: "we trained computational models by machine learning algorithm
[sic] for predicting SPS scores based on suicide dictionary [sic].
Meanwhile, we also build [sic] similar computational models using
LIWC."

The authors also need to tell us what LIWC features they used in their
model. Was it all ~80 LIWC dimensions or was it just the handlful of
LIWC features identified in e.g. Stirman and Pennebaker? Table 2
should include the LIWC features (or some relevant subset if they used
all 80) correlations in addition to their suicide dictionary dimension
ratings.

I have some concerns about whether the article is within the scope of
Peer J Computer Science. In my mind, the paper's main focus is the
development and validation of a new dictionary of suicide-risk
relevant Chinese-language terms. In this way, it may be more appropriate to a less CS-focused journal such as one focused on corpora or perhaps a journal focused on social media. The paper does not have much in the way of novel algorithms or computational methods. Yes, the author's do use word2vec to expand their corpus. But the words resulting from this method are still passed to humans to judge if they are really good words to add to the suicide dictionary. A paper with more of a focus on novel algorithms and methods would be more in line with the scope of Peer J Computer Science.

Validity of the findings

Another major problem with the article is that I don't think they
validly can support their conclusion that "the suicide dictionary
performs as efficiently as human coding in [a] detecting suicidal
expression [sic] and [b] identifying one's level of suicide risk".

They are stating that they can do two things with their results,
i.e. [a] and [b] that I inserted in the above quote. At best they can
do one thing. In particular, their correlations in Table 2 and the
precision/recall figures in Table 3 appear to be at the level of
individual people and *not* at the level of individual postings
(i.e. 'suicidal expressions'). Thus, [a] can not be proven from the
results given - in fact I did not see any analysis at the level of
individual expressions. That said, Table 2 and 3 *are* relevant to
[b] because the data are at the level of individual people.

However, even claims of the form of [b] are over-stated. For example,
on line 322 to 324 it is asserted that "[sic] suicide dictionary,
performed as efficiently as expert [sic]." I do not see how this is
proven. Sure, the correlation was 0.455. That does not mean that the
performance of dictionary is as good as the experts.

The authors have divided the sample of Weibo people into three groups,
a high risk group (with higher than average SPS scores) and low risk
group (with lower than average SPS scores) and a middle risk group.
They then remove the middle risk group from their analysis. This is
an artificial and extreme simplification of the real problem. The
real problem is to take an arbtirary persons's Weibo posts and decide
are they high risk or not. Thus the two groups should be (1) high
risk and (2) medium to low risk.

It might be okay to simplify a classification problem if, for example,
one is trying to show that one scheme is better than another. The
authors would be on firmer ground if that was their explicitly stated
goal. However, this is not the author's claim. The author's claim is
that the scheme is "as good as human performance."

It is puzzling to me what role the 31 suicidal users postings had in
the study. Apparently their postings had only a minor role. I kept
hoping to see that sample provided as some kind of test set -- were
the human experts able to pick out these individuals postings? How
did the dictionary rate these users's postings? As best I could tell,
these users postings were merely used along with the rest of the
postings to serve as a basis for finding words relevant to suicide.
The high vs low risk SPS groups were used as a gold standard. SPS is
only an indirect measure of suicide risk. The 31 suicidal posters is
a better set of "gold" labeled data. To make the study more valid,
the authors should show that their methods can predict the suicidal
risk of these posters.

Additional comments

The authors take on a very important problem.

The dataset seems like it can be a very valuable resource. (However, minor detail: I would really like to have seen what is the min, max, mean, median length in words of Weibo posts).

The authors may consider a workshop or a conference venue for this work. The barrier to entry will be lower. Additionally, there may be a conference or workshop that focuses on corpora. The authors should consider a computational linguistics / clinical psychology oriented workshop such as CLPsych (e.g. http://clpsych.org/).

---

## Round 0.2 · Minor Revisions

· Academic Editor

Minor Revisions

I would like to thank the authors for their patience and perseverance. Although 1 reviewer suggested acceptance, the other reviewer (Reviewer 3) raised significant concerns that, if addressed, would significantly improve the paper. As these changes involve the presentation and explanation of the methods and result, I believe they can be easily addressed, and the paper would be much improved as a result. Please respond to reviewer 3's concerns. I look forward to seeing your revised manuscript.

Reviewer 1 ·

Basic reporting

None.

Experimental design

None.

Validity of the findings

None.

Additional comments

The manuscript has been revised well. I think this manuscript is acceptable.

Reviewer 3 ·

Basic reporting

* Abstract:

In the abstract there is a Methods section. The third sentence in
their is confusing. It sounds like what they are trying to say is
that they wanted to predict an individual's level of suicide risk by
automatically analyzing their tweets. They used two gold standards
for making this prediction, (a) a self-report mesaure called SPS and
(b) expert's ratings. They developed two such predictors. One such
predictor used linguistic features from a pre-existing dictionary
called CLIWC. They other used linguistic features from a dictionary
that the authors created. The compared the performance of each of the
two predictors against the gold standards (a) and (b).

Do I understand things correctly?

Please re-write the method section in the abstract to minimize the
confusion. As I said, the third sentence in particular really needs
clarification.



* Review and Discussion Section

The Review and Discussion section is also quite unclear.

They report two correlations. What does each correlation measure?
They report 4 different F measures. One pair of F's is for the
predictor derived from their suicide dictionary. The other pair of
F's are from a predictor that used CLIWC for its features. But what
are the two Fs for each predictor? Are they from two different
observation times? If so, maybe say something like "t1: F1 = 0.48,
t2: 0.56).

It is also unclear because in the Abstract's Mini-Method section I thought
they mentioned there were two gold standards. Not just SPS, but also
expert ratings. Why don't I see any mention of expert ratings in this
Abstract Results and Discussion section???

* Reference Section

I have some concerns about the references

First there was this regarding the SPS:

Cull and Gill 1989. <etc>

...I have been unable to find that online. I was able to find this...

Cull JG, Gill WS: Suicide Probability Scale (SPS) Manual. Los Angeles,
CA, Western Psychological Services, 1986

....The authors should double check this (was it really 1989) and let
us know what is going on.

In addition they say they used a Chinese version of the SPS. They
cite Huang 2012 "The Chinese version of SPS was used in this study
(Huang et al., 2012)" Is this *really* true? I do not have access to
this article. The title of that article is "The development of the
Chinese Linguistic Inquiry and Word Count Dictionary". Given that
title it seems exceedingly unlikely that this article is about the
Chinese SPS.

As I say elsewhere, they currently cite Pennebaker 2001 in the
reference section but do not mention him in the text. That said it is
a minor point (especially because I suggest a reason for them to cite
it directly)

Experimental design

* Introduction

I found this paper...

"Detecting Suicidal Ideation in Chinese Microblogs
with Psychological Lexicons" by Huang et al.
http://arxiv.org/pdf/1411.0778v1.pdf.

...the authors really need to cite it. It, like the present paper
tries to predict suicide risk from Sina Weibo (which is like the
Chinese version of twitter).

Given what seems to be a strong overlap, the authors should clarify what is original in their submission.

Validity of the findings

* Discussion Section (two Validity of Findings issues)

Table 2: Why are there three observation windows in Table 2 and only
two observation windows mentioned in the abstract and in Table 3.
This seems like a glaring inconsistency and/or omission that needs to be clarified.

Table 3: Predicting SPS scores using suicide dictionary and CLIWC

Another thing that really baffles me is how can the authors
report precision/recall and F measure on predicting SPS? SPS produces
a 3 summary scores and 4 subclinical scores (SOURCE:
http://www.sprc.org/sites/sprc.org/files/library/BrownReviewAssessmentMeasuresAdultsOlderAdults.pdf).

After lots of jumping back and forth I think I see what they did based
on this quote: "All participants were divided into high risk and
non-high (medium to low) risk group based on thresholds (mean value ±
standard deviation)."

The authors must more clearly state throughout that there were these
two groups. The groups were based on SPS score. Table 3 is *not*
"Predicting SPS" as the title of table 3 currently indicates. Table 3
is for predicting group. The title of Table 3 must be changed
accordingly.

The authors must state what the mean and sd were for SPS and what the
thresholds used were.

The authors should make some bold heading in method divindng the SPS
from the expert ratings section etc.

* An Additional Limitation to Report In Discussion

It was thoughtful of the authors to list some limitations of their
study. They should mention an additional limitation because it may
stimulate important new work.

The limitation is that current word-category approach does not take
contextual factors into account. This is because each dictionary
feature vector represents the frequency of words in a particular
word-category. Depression and suicide were found to be associated
with higher frequency of words in LIWC's self-focused word
category. This category includes words like "I", "me", "my", "I'd",
etc). However, other studies have not always found this association
to hold up. The association has been shown to be dependent on the
surrounding linguistic context Jarrold et al (2012).

For this reason, future work should look not just at the frequency of
single words (or more precisely the frequency of single words of a
particular word type). Rather it should look at words in context.
One such means of looking at the broader context may be involve n-gram
features (bigrams, tri-grams etc).

I notice that the authors site Stirman and Pennebaker in the reference
section but not in the text.

Here is the citation for Rude(2004):

Rude, S. S., Gortner, E. M., & Pennebaker, J. W. (2004) Language use
of depressed and depression-vulnerable college students. Cognition and
emotion, 18, 1121-1133.

Here is the citation for Jarrold(2012):

Jarrold, W., Javitz, H.S., Krasnow, R., Peintner, B., Yeh E., Swan,
G.E. (2011) Depression and Self-Focused Language in Structured
Interviews with Older Adults Psychological Reports
Oct;109(2):686- 700.

Additional comments

The writing has definitely come a long way and suicide risk detection remains
an important topic.

However, after reading this article I still don't have a clear sense of whether
they tried to classify tweets or individuals. The article needs to be
re-written to make this clear from the abstract throughout all the
text and the conclusion.

It is a sign that there are still issues with basic organization and clarity.

---

## Round 0.3 · Minor Revisions

· Academic Editor

Minor Revisions

Thank you for your revision. I would like to thank the reviewer, who has carefully reviewed this submission and provided substantive comments that would, if addressed, improve the readability and utility of this paper. Please address the changes as identified.

Reviewer 3 ·

Basic reporting

Please see "General Comments for the Author"

Experimental design

Please see "General Comments for the Author"

Validity of the findings

Please see "General Comments for the Author"

Additional comments

I appreciate the effort and tenacity of the authors in responding to
two reviews. The article has come a long way and is of higher
scientific quality. The majority of my critiques have been responded
to.

Unfortunately there still remain some somewhat significant issues but
less glaring than before -- see Section 1. In addition, now that I
can understand more of the paper, I have other new critiques that
should be addressed -- see Section 2.

* Section 1: Responses to the Re-Submission Letter

There were 16 points in the re-submission letter. I am reasonably
happy with all of them except for two places as I describe below.

I refer to these points based on the number given in their
re-submission letter.

** Point 10 Of Resubmission Letter

With respect to point number 10 in their re-submission letter The
authors have made progress but there are still problems.

The fundamental problem is that the authors are over-stating what they
have done.

Let me be specific:

This phrase in the Abstract...

"For predicting individual SPS scores, the Chinese suicide dictionary
(t1: F1=0.48; t2: F1=0.56)"

...still makes it sound like you are trying to predict the SPS score.

You can't possibly use an F1 to measure the quality of predicting the
SPS score. The SPS produces a number as a score. An F1 score is a
statistical analysis of *binary* classification - it can not tell
anything about how well it predicts a score.

It is much harder to predict an actual score than to make a binary
classification. By claiming that you are predicting the score you are
overstating what you really did.

The problem is still manifest in other places.

For example, like the sentence in the abstract, the title of Table 3
needs to be clarified. For example they need to change it from
"Predicting individual scores on SPS..." to something like "Predicting
High vs Non-High SPS Group..." or even "Predicting SPS Group"
providing that you clearly define SPS Group elsewhere in the article.

Another place I have found the problem:

Under Data Analysis section part "c" the is the phrase "For testing
the accuracy in predicting individual scores on self-rating measure of
suicide risk". Again, this is the same problem in over claiming what
you have done. Please make it crystal clear that you are merely
doing a binary classification than a prediction of a score.

Please ensure there are no other places in the text where you are
making this overstatement.

** Regarding Point 13, 14 and 15 in the Resubmission Letter.

This section collectively covers Point 13, 14 and 15 in the letter.

Thank you for adding the reference to Rude 2004 and Jarrold 2011. It
is a pity that you removed the text that contained Stirman and
Pennebaker reference but I won't raise a big fuss.

Finally,I stand corrected, In my prior critique I said mentioned Stirman and Pennebaker 2001 in the Reference section without actually mentioning them in the text of their submission. The authors correctly pointed out that they did indeed refer to this article in the their text. My apologies.

However, if you could please cite Jarrold 2011 et al after this
sentence in the Discussion section...

". However, changes in elements of surrounding linguistic context may
determine the relationship between individual psychological features
and patterns of language use."

....that would be much better. After all, as I said in my prior
critique, a main focus of that article is the context-dependence of
language features. It is ok to cite it as you have done in the
background to support "The words that people use provide important
psychological cues to their mental health status" however, it is more
important to cite it as I suggest in the Discussion section.

It is odd to me that you begin that sentence with the word "However".
That sentence is consistent and in support of the prior sentence thus
why does it begin with "however". A nitpick for sure, but one of many
signs that the writing can still be improved.

* Section 2: New Critiques

Now that I can understand more of the paper I have additional
critiques. Please find them immediately below.

** Point A: Emphasize Multi-dimensional Nature of Dictionary

Table 2. I thought a main point of the paper was to show how their
hand constructed dictionary is better than CLIWC. If so, why did the
authors not show the correlations of CLIWC with their new suicide
dictionary?

I did a lot more reading and rummaging around in the text and figured
it out, it is because their new dictionary has a bunch of categories
that CLIWC does not have.

(Thus the question in the preceding paragraph can be ignored as a critique. I am just pointing out how confused a reader can get and how much guidance and re-guidance you should continually be giving them).

This made me realize that the authors need to be more clear about the
multidimensional nature of their new dictionary.

For example, if they could change this sentence...

"All three coders were required to rate how each Weibo post relates to
each one category defined in the Chinese suicide"

...to....

"All three coders were required to rate how each Weibo post relates to
each one OF THE XXX CATEGORIES defined in the Chinese suicide"

....and if they could change...

"Finally, the suicide dictionary is composed of 2,168 words
(1,862+306=2,168)."

...to...

"Finally, the suicide dictionary is composed of 2,168 words
(1,862+306=2,168) SPREAD AMONGST XXX CATEGORIES"

...that would do a lot to prevent readers from being confused.

(Obviously, XXX refers to however many categories they are using)

There are probably other places where they should be clearer about the
number of categories.

** Point B: Another Point on Readability:

Make "Data Analysis" on par with the "Measurement" section. That is,
"Data Analysis" like "Measurement" would be set off on its own line.

Additionally (and more importantly) give each of the three sub-part of
"Data Analysis" (i.e. (a), (b) and (c)) its own heading, preferably in
bold such that they have the same "status" as "Expert Ratings" and
"Suicide Possibility Scale"

** Point C: A Comment on the overall Design:

It is now clear to me that the weights of each feature were
hand-assigned. You would probably have much better accuracy in
predicting based off of individual tweets if you computed the weights
rather than had humans guess them. It is up to you whether you want
to do that for this article -- obviously a lot more work -- , propose
it as future work or ignore this comment.

* CONCLUSION

In sum I hope you find these comments helpful. A clearly written paper will be a paper that is better understood and thus more likely to be cited.

---

## Round 0.4 · accepted · Accept

· Academic Editor

Accept

Thank you very much for your careful responses to the reviewer's concerns, which I believe that you have addressed appropriately.